# Linear Radical Additions-Coupling Polymerization (LRAsCP): Model, Experiment and Application

**DOI:** 10.3390/polym17060741

**Published:** 2025-03-12

**Authors:** Yudian Jiang, Kun Cao, Qi Wang

**Affiliations:** 1MOE Key Laboratory of Macromolecular Synthesis and Functionalization, Department of Polymer Science & Engineering, Zhejiang University, Hangzhou 310058, China; 22029005@zju.edu.cn; 2State Key Laboratory of Chemical Engineering, College of Chemical and Biological Engineering, Zhejiang University, Hangzhou 310058, China; kcao@che.zju.edu.cn

**Keywords:** linear radical additions-coupling polymerization (LRAsCP), kinetics analysis, bifunctional initiator, multiblock (co)polymer

## Abstract

Exploring new polymerization strategies for currently available monomers is a challenge in polymer science. Herein, a bifunctional initiator (BFI) is introduced for the conventional radical polymerization of a vinyl monomer, resulting in linear radical additions-coupling polymerization (LRAsCP). In LRAsCP, the coupling reaction alongside the addition reaction of the radicals contributes to the construction of polymer chains, which leads to stepwise growth of the multiblock structure. Theoretical analysis of LRAsCP predicted variation of some structural parameters of the resulting multiblock polymer (MBP) with the extent of initiation of the BFI and the termination factor of the radicals. Simultaneous and cascade initiations of the BFI were compared. LRAsCP of styrene was conducted, and a kinetics study was carried out. The increment in *M*_n_ with polymerization time demonstrated the stepwise mechanism of the formation of the MBP. The variation of the structural parameters of MBP fitted well with the theoretical prediction. Two-step LRAsCP was conducted and multiblock copolymers (MBcP) were obtained either by in situ copolymerization of styrene and MMA or by a second copolymerization of styrene and BMA. The current results demonstrate that the introduction of a BFI to conventional radical polymerization generates a new polymerization strategy, leading to a new chain architecture, which can be extended to other radical polymerizable monomers.

## 1. Introduction

Multiblock polymer (MBP) has extensive applications in various fields [1,2,3,4,5]. Currently, its synthesis strategy includes three approaches. One is the construction of the polymer chain ‘block by block’ via sequential living polymerization [6,7,8,9,10,11] (Figure 1A(1)), such as anionic polymerization [12,13], controlled radical polymerization [8,9,14,15,16,17,18,19,20,21] and olefin metathesis polymerization [10]. Although well-defined MBPs can be prepared under critical polymerization conditions with multiple steps, the tolerance of anionic polymerization limits its application in monomers with active groups, such as carboxyl and hydroxyl groups. While controlled radical polymerization can be applied to more types of monomers, the low polymerization rate implies that achieving the designed conversion and the molar mass of the polymer requires a long time to reach the designed conversion and molar mass of the polymer. The second approach is the coupling of α, ω-telechelic polymers prepared by living polymerization, which needs more than two steps (Figure 1A(2)) [22,23,24,25,26,27]. Due to the relatively low activity of the terminal group of macromolecules, the efficiency of the coupling reaction is unsatisfactory. The third approach is the combination of different polymerization mechanisms, which meets the same problems existing in individual methods. Exploring simple and easy methods for building polymers with complex segmental structures is still a challenge in polymer synthesis.

Radical coupling reactions are rapid reactions that have been applied in some polymerizations as a tool for building carbon–carbon bonds. Direct coupling of biradicals generates linear polymer chains (Figure 1B(1)) [28,29,30,31,32]. The reaction between biradicals and various non-homopolymerized double bonds results in multi-segmental or multiblock linear polymers (Figure 1B(2)) [22,23,24,25,33,34,35,36,37]. When biradicals or polyradicals are involved in radical polymerization of vinyl monomers, the radical coupling reaction is a chain extension step instead of a termination step. Radical polymerization of monovinyl monomers initiated by a polyfunctional initiator (PFI) leads to the formation of a branched polymer or polymer network. This strategy is termed non-linear radical additions-coupling polymerization (NLRAsCP) [38], in which the PFI is the source of the branching. The mechanism of NLRAsCP demonstrates that gelation is determined by the extent of initiation of the functional groups of the initiator (*q*) and the coupling factor of the macroradical (*ϕ*).

Radical polymerization is one of the oldest polymerization methods and tolerates various functional groups. It can be anticipated that when a bifunctional initiator (BFI) is introduced in the polymerization of monovinyl monomers, the coupling reaction along with the addition reaction of the radical contributes to the construction of polymer chains, producing a linear polymer composed of a multiblock structure if chain transfer to the polymer is absent. This polymerization is termed linear radical additions-coupling polymerization (LRAsCP, Figure 1A(3)).

In this paper, we propose an efficient protocol to construct a multiblock structure via one-step LRAsCP, and two different blocks via two-step LRAsCP. The kinetic analysis of LRAsCP and its application in the synthesis of MBP and multiblock copolymers (MBcP) are investigated. Moreover, LRAsCP provides a simple approach for determining the contribution of the coupling reaction to the termination step in radical polymerization.

## 2. Theoretical Analysis of LRAsCP

### 2.1. General Process of LRAsCP

The process of LRAsCP is shown in Figure 2. It includes three steps, such as initiation, propagation and coupling/termination, similar to normal radical polymerization. In the initiation step, the carbon radicals are formed by radical transfer reaction between bromides and pentacarbonyl manganese radical (Mn(CO)_5_•), which is generated by homolysis photolysis of decacarbonyl dimanganese (Mn_2_(CO)_10_) under visible light (see Appendix A) [39,40,41]. I_1_XI_2_, the BFI, generates three kinds of small radicals, such as I_2_XI_1_•, I_1_XI_2_• and •I_1_XI_2_•. The first two radicals are monoradical, and the third one is biradical. In the propagation step, the three small radicals generate three macroradicals through the radical addition reaction with monomer A, such as I_2_XA_j_•, I_1_XA_j_• and •A_k_XA_j_•. The number of macroradicals is the same as that of the corresponding small radicals. In the coupling/termination step, the radical coupling reaction between two radicals is a chain extension step instead of the termination step in conventional radical polymerization initiated by monoradical. Other reactions, such as disproportionation and chain transfer reaction, are termination steps which result in a dead end of unit A. If radicals are considered as functional groups, the coupling reaction between the macroradicals is a kind of linear coupling polymerization.

Let *q*_1_ and *q*_2_ be the extent of initiation of groups I_1_ and I_2_ of BFI, and *q* be the total extent of initiation of BFI; then, *q* is equal to 0.5(*q*_1_ + *q*_2_). Let *ϕ*_A_ be the termination factor involving radical A, which is the ratio of the rate of radical coupling reaction to the overall rate of reaction consuming radicals [38]. Let *δ*_11_, *δ*_12_, *δ*_2_ and *δ*_0_ be the molar fraction of small radicals I_2_XI_1_•, I_1_XI_2_•, •I_1_XI_2_• and free BFI, respectively. The macroradicals, I_2_XA_j_•, I_1_XA_j_• and •A_k_XA_j_•, are formed via the radical addition reaction of the corresponding small radicals to monomer A. Their molar fractions are equal to *δ*_11_, *δ*_12_ and *δ*_2_, in terms of molecule, respectively. With respect to radicals, the molar fractions of the three macroradicals are represented by *θ*_11_, *θ*_12_ and *θ*_2_. The parameters of the *δ*- and *θ*-series can be represented by *q*_1_ and *q*_2_, which are summarized in Table 1.

The coupling reaction between three macroradicals generates six MBPs, which can be categorized into three types according to their terminal groups shown in Figure 2. The first type is a polymer with two terminal units A, denoted as AA. The second type is a polymer with one functional group I_1_ or I_2_ and one terminal unit A, denoted as I_1_A or I_2_A. Both are collectively denoted as IA. The third type is a polymer with two functional groups, I_1_ or I_2_, denoted as I_1_AI_1_, I_1_AI_2_ and I_2_AI_2_. All three are collectively denoted as IAI.

### 2.2. The Number-Distribution Functions of MBP

To discuss the kinetics of LRAsCP, the following assumptions should be accepted. (1) The reactivity of all radicals is equivalent. (2) The intramolecular cycling reaction is ignored. (3) The contribution of additional monoradical generated by chain transfer is neglected. Based on the above assumptions, the number-distribution functions (*p*_MBP, *n*_) of the six MBPs are given by (1)$$p_{AA,n}=\delta_{2}{\left(\theta_{2}\phi_{A}\right)}^{n-1}{\bar{\phi_{A}}}^{2}={q_{1}q_{2}{\bar{\phi_{A}}}^{2}(\frac{2q_{1}q_{2}}{q_{1}+q_{2}}\phi_{A})}^{n-1}\;\;\;\;\;\;\;\;\;\;\;\;\;\;\;\;(n\geq 1)$$
(2)$$p_{I_{1}A,n}=\delta_{12}{\left(\theta_{2}\phi_{A}\right)}^{n-1}\bar{\phi_{A}}={\bar{q_{1}}q}_{2}\bar{\phi_{A}}{\left(\frac{2q_{1}q_{2}}{q_{1}+q_{2}}\phi_{A}\right)}^{n-1}\;\;\;\;\;\;\;\;\;\;\;\;\;\;\;\;(n\geq 1)$$
(3)$$p_{I_{2}A,n}=\delta_{11}{\left(\theta_{2}\phi_{A}\right)}^{n-1}\bar{\phi_{A}}={\bar{q_{2}}q}_{1}\bar{\phi_{A}}{\left(\frac{2q_{1}q_{2}}{q_{1}+q_{2}}\phi_{A}\right)}^{n-1}\;\;\;\;\;\;\;\;\;\;\;\;\;\;\;\;(n\geq 1)$$
(4)$$p_{I_{1}AI_{1},n}=0.5\delta_{12}{\left(\theta_{2}\phi_{A}\right)}^{n-2}\theta_{12}\phi_{A}=0.5\frac{{\left(q_{2}\bar{q_{1}}\right)}^{2}}{q_{1}+q_{2}}\phi_{A}{\left(\frac{2q_{1}q_{2}}{q_{1}+q_{2}}\phi_{A}\right)}^{n-2}\;\;\;\;\;\;\;\;\;\;\;(n\geq 2)$$
(5)$$p_{I_{1}AI_{2},n}=\delta_{12}{\left(\theta_{2}\phi_{A}\right)}^{n-2}\theta_{11}\phi_{A}=\frac{q_{1}q_{2}\bar{q_{1}}\bar{q_{2}}}{q_{1}+q_{2}}\phi_{A}{\left(\frac{2q_{1}q_{2}}{q_{1}+q_{2}}\phi_{A}\right)}^{n-2}\;\;\;\;\;\;\;\;\;\;\;\;\;(n\geq 2)$$
(6)$$p_{I_{2}AI_{2},n}=0.5\delta_{11}{\left(\theta_{2}\phi_{A}\right)}^{n-2}\theta_{11}\phi_{A}=0.5\frac{{\left(q_{1}\bar{q_{2}}\right)}^{2}}{q_{1}+q_{2}}\phi_{A}{\left(\frac{2q_{1}q_{2}}{q_{1}+q_{2}}\phi_{A}\right)}^{n-2}\;\;\;\;\;\;\;\;\;\;\;(n\geq 2)$$
where *n* is the number of BFI or X (residual moiety of BFI).

For example, the coupling reaction of *n* biradical •A_k_XA_j_• results in the formation of AA-type of MBP containing *n* X units. The formation of this chain starts with one •A_k_XA_j_•, followed by (*n* − 1) coupling reactions of the same biradical and two non-coupling reactions of two terminal radicals. The corresponding probabilities for the three steps are $\delta_{2}$, ${\left(\theta_{2}\phi_{A}\right)}^{n-1}$ and ${\bar{\phi_{A}}}^{2}$. The product of the three probabilities yields Equation (1). The formation of I_1_AI_2_-type MBP starts with an I_1_XI_2_• radical, followed by coupling reaction with (*n −* 2) •A_k_XA_j_• biradicals and an I_2_XA_j_• radical. The product of the corresponding probabilities, $\delta_{12}$, ${\left(\theta_{2}\phi_{A}\right)}^{n-2}$ and *θ*_11_*ϕ*_A_, yields Equation (5). Other distribution functions can be derived by the same method and are given in Equations (1)–(6). The factor of 0.5 in Equations (4)–(6) is due to the symmetry of I_1_AI_1_- and I_2_AI_2_-type MBPs. The theoretical analysis of kinetics is described in the Appendix A in detail. According to Equations (1)–(6), the distribution functions depend on *q*, *ϕ*_A_ and *n*.

### 2.3. The Structural Parameters of MBP

The structural parameters of MBP can be calculated from its number-distribution functions shown in Equations (1)–(6), such as the number and the fraction of each MBP (*N* and *F*) and its number-average degree of polymerization (*DP*_n_) in terms of X. *N*, *DP*_n_, the number-average molar mass (*M*_n_) and the average number of blocks per MBP (*N*_K_) of total MBPs can be obtained as well. Except *M*_n_, all structural parameters of MBP are determined by three factors, *q*_1_, *q*_2_ and *ϕ*.

When more than one kind of monomer participate in LRAsCP, multiblock copolymer (MBcP) can be obtained, which is termed linear radical additions-coupling copolymerization (LRAsCcP). On the other side, at the polymerization time when the extent of initiation of I_1_ and I_2_ are *q*_10_ and *q*_20_, the addition of monomer B to the polymerization mixture leads to the second-step polymerization resulting in copolymer block. Two-step LRAsCP generates two different blocks. The structural parameters of MBcP can be derived from its number-distribution function as well, which is described in the SM in detail.

## 3. Discussion on Theoretical Predictions of LRAsCP

Based on the mechanism of LRAsCP, the multiblock structure is predicted to be formed during the polymerization. We briefly discuss some important aspects of LRAsCP of typical monomers, such as styrene and methyl methacrylate (MMA) with *ϕ*_A_ values of 0.93 and 0.33 [38].

### 3.1. Numbers and Fractions of MBPs (N_MBP_ and F_MBP_)


(7)
$$N_{AA}=q_{1}q_{2}{\bar{\phi_{A}}}^{2}\frac{q_{1}+q_{2}}{q_{1}+q_{2}-2q_{1}q_{2}\phi_{A}}$$



(8)
$$N_{IA}=N_{I_{1}A}+N_{I_{2}A}=\frac{(q_{1}+q_{2})(q_{1}+q_{2}-2q_{1}q_{2})\bar{\phi_{A}}}{q_{1}+q_{2}-2q_{1}q_{2}\phi_{A}}$$



(9)
$$N_{IAI}=N_{I_{1}AI_{1}}+N_{I_{1}AI_{2}}+N_{I_{2}AI_{2}}=\frac{{0.5\phi_{A}(q_{1}+q_{2}-2q_{1}q_{2})}^{2}}{q_{1}+q_{2}-2q_{1}q_{2}\phi_{A}}$$



(10)
$$N_{P_{A}}=N_{AA}+N_{IA}+N_{IAI}=\left(q_{1}+q_{2}\right)\left(1-{0.5\phi }_{A}\right)-q_{1}q_{2}$$


The numbers of MBPs (*N*) can be calculated by Equations (7)–(10) assuming the initial number of BFI (X_0_) is unity, which suggests that *N* depends on *q*_1_, *q*_2_ and *ϕ*_A_. Taking *ϕ*_A_ = 0.93 [38] as an example, the variation tendency of *N* is shown in Figure 1. The number of AA-type MBP (*N*_AA_) monotonically increases with both *q*_1_ and *q*_2_ and increases rapidly as *q* is close to unity. (Figure 1A) The number of IA-type MBP (*N*_IA_) quickly increases with both *q*_1_ and *q*_2_ at the beginning of polymerization. (Figure 1B) After it reaches the maximum, it rapidly decreases to null upon complete initiation. The number of IAI-type MBP (*N*_IAI_) increases with *q* and decreases after it reaches the maximum. (Figure 1C) Moreover, its variation depends on the relative activity of the two functional groups of BFI. When the activities of I_1_ and I_2_ are equal (*q*_1_ = *q*_2_ = *q*), the case is termed simultaneous-initiation (*s*-initiation). When the activities of the two groups are remarkably different and the reactivity of I_1_ is much higher than I_2_, the case is termed cascade-initiation (*c*-initiation). Two special cases are shown in each figure by black and red lines, respectively. As shown by the black line in Figure 1C, *N*_IAI_ derived from *s*-initiation case is lower than other cases (*q*_1_ ≠ *q*_2_). As shown by the red line in Figure 1C, *N*_IAI_ derived from *c*-initiation case is larger than other cases. The variation tendency of the number of total MBPs (*N*_PA_) shown in Figure 1D is similar to that of *N*_IAI_.(11)$$F_{AA}=\frac{N_{AA}}{N_{P_{A}}}=\frac{(q_{1}+q_{2})q_{1}q_{2}{\bar{\phi_{A}}}^{2}}{\left(q_{1}+q_{2}-2q_{1}q_{2}\phi_{A})(\left(q_{1}+q_{2}\right)\left(1-{0.5\phi }_{A}\right)-q_{1}q_{2}\right)}$$(12)$$F_{IA}=\frac{N_{IA}}{N_{P_{A}}}=\frac{(q_{1}+q_{2})(q_{1}+q_{2}-2q_{1}q_{2})\bar{\phi_{A}}}{\left(q_{1}+q_{2}-2q_{1}q_{2}\phi_{A})(\left(q_{1}+q_{2}\right)\left(1-{0.5\phi }_{A}\right)-q_{1}q_{2}\right)}$$(13)$$F_{IAI}=\frac{N_{IAI}}{N_{P_{A}}}=\frac{{0.5\phi_{A}(q_{1}+q_{2}-2q_{1}q_{2})}^{2}}{\left(q_{1}+q_{2}-2q_{1}q_{2}\phi_{A})(\left(q_{1}+q_{2}\right)\left(1-{0.5\phi }_{A}\right)-q_{1}q_{2}\right)}$$

The molar fractions (*F*) of three MBPs are given by Equations (11)–(13), and their variation tendency is shown in Figure 2. Two typical cases, such as *ϕ*_A_ = 0.93 for styrene and 0.33 for MMA [38], were compared. It can be found in Figure 2A1 and A2 that the molar fraction of AA-type MBP (*F*_AA_) is very low at the beginning of polymerization and increases gradually with *q* for both cases. When *q* is close to unity, *F*_AA_ increases rapidly. As shown in Figure 2C1 and C2, the variation tendency of molar fraction of IAI-type of MBP (*F*_IAI_) is opposite to *F*_AA_ for both cases. The difference between the two cases is *F*_IAI_ for styrene is much higher than that of MMA, which is due to the larger *ϕ*. The molar fraction of IA-type MBP (*F*_IA_) for styrene increases with *q* and reaches the maximum before the end of the initiation. (Figure 2B1) After it reaches the maximum, it decreases to null upon complete initiation. For MMA, *F*_IA_ decreases with *q* and reaches null upon complete initiation. (Figure 2B2) The difference between the two cases is mainly due to the different values of *ϕ*.

### 3.2. The Number-Average Degree of Polymerization in Terms of X (DP_n_)


(14)
$${DP}_{n,AA}={DP}_{n,IA}=\frac{q_{1}+q_{2}}{q_{1}+q_{2}-2q_{1}q_{2}\phi_{A}}$$



(15)
$${DP}_{n,IAI}=2\frac{q_{1}+q_{2}-q_{1}q_{2}\phi_{A}}{q_{1}+q_{2}-2q_{1}q_{2}\phi_{A}}={DP}_{n,IA}+1$$



(16)
$${DP}_{n}=\frac{q_{1}+q_{2}-q_{1}q_{2}}{\left(q_{1}+q_{2}\right)\left(1-0.5\phi_{A}\right)-q_{1}q_{2}}$$


The chain length of various MBPs formed in LRAsCP can be quantitatively evaluated by the incorporated residual BFI (X). *DP*_n_ in terms of X of various MBPs and their sum can be calculated by Equations (14)–(16), and the variation tendency with *q* is shown in Figure 3. It can be concluded that *DP*_n_ increases with both *q*_1_ and *q*_2_ and increases rapidly when both *q*_1_ and *q*_2_ approach unity. This tendency is similar to the dependence of the degree of polymerization on polymerization time or the extent of reaction in stepwise polymerization [42]. When *q* exceeds 0.5 or both functional groups are initiated, the *DP*_n_ of total MBPs gradually increases, and rapidly increases when *q* is close to unity. The variation tendency of *DP*_n_ suggests that LRAsCP follows the stepwise mechanism since the coupling reaction is the key step in the construction of the multiblock architecture.

When *s*-initiation and *c*-initiation cases are compared, the difference of *DP*_n_ can be found when *q* is less than 0.5. As shown in Figure 3D, *DP*_n_ increases gradually with *q* and increases rapidly when *q* is close to unity for *s*-initiation case (black line), while it remains constant when *q* is less than 0.5 or *q*_2_ = 0 for *c*-initiation case (red line). This is due to the selective initiation of one functional group of BFIs and the formation of monoradical in the early stage of *c*-initiation case (*q* < 0.5), which is the same as radical polymerization initiated by monofunctional initiator (MFI). *q* exceeds 0.5 when the second functional group of each BFI is initiated. Initiation of two functional groups results in the continuous formation of MBPs and an increment of *DP*_n_ in *c*-initiation case.

### 3.3. The Number-Average Molar Mass of Total MBPs (M_n_)

Based on the number of all MBPs ($N_{P_{A}}$), the monomer conversion (*C*) and the number-average molar mass of total MBPs (*M*_n_) can be calculated by Equation (17).(17)$$M_{n}=\frac{A_{0}Cm_{A}}{(\left(q_{1}+q_{2}\right)\left(1-0.5\phi_{A}\right)-q_{1}q_{2})X_{0}}$$
where *A*_0_, *X*_0_, *C*, and *m*_A_ are the initial concentration of monomer A and BFI, the monomer conversion and the molar mass of monomer, respectively. Since the *M*_n_ of MBP is determined by the feed ratio and the monomer conversion in addition to *q*_1_, *q*_2_ and *ϕ*_A_, the discussion will be given in experimental studies.

### 3.4. The Average Number of Blocks per MBP (N_K_)

The average number of blocks per MBP (*N*_K_) can be evaluated by the total number of blocks and $N_{P_{A}}$, which is given by Equation (18).(18)$$N_{K}=\frac{\left(q_{1}+q_{2}\right)\left(1-0.5\phi_{A}\right)}{\left(q_{1}+q_{2}\right)\left(1-0.5\phi_{A}\right)-q_{1}q_{2}}$$

This parameter can be experimentally evaluated and will be further discussed in experimental studies.

### 3.5. The Second-Step Polymerization

#### 3.5.1. Free BFI and AA-Type MBP


(19)
$$F_{BFI}=\delta_{0}=\left(1-q_{1}\right)\left(1-q_{2}\right)$$



(20)
$$W_{AA}=q_{1}q_{2}{\bar{\phi_{A}}}^{2}{\left(\frac{q_{1}+q_{2}}{q_{1}+q_{2}-2q_{1}q_{2}\phi_{A}}\right)}^{2}$$


The molar fractions of free BFI and AA-type MBP are two parameters that need to be considered if the second LRAsCP of monomer B is conducted. Free BFI might generate BB-type MBP in the second polymerization, while AA-type MBP will not initiate the second polymerization of monomer B and cannot be incorporated into MBcP. Both AA- and BB-type MBPs are impurities with respect to MBcP.

The molar fraction of free BFI (*F*_BFI_) is derived in Equation (19), and its variation with *q* is shown in Figure 4A. When the total extent of initiation (*q*) is fixed, *F*_BFI_ has the maximum value when *q*_1_ and *q*_2_ are equal, and the minimum value when the difference between *q*_1_ and *q*_2_ is the largest. For example, free BFI disappears when *q*_1_ = 1 and *q*_2_ = 0 or *q* ≥ 0.5 for *c*-initiation case (red line in Figure 4A). When *q*_1_ = *q*_2_ = *q* = 0.5, 25% of BFI remains for *s*-initiation case (black line in Figure 4A). Therefore, *c*-initiation case has the lowest value of *F*_BFI_. In other words, *c*-initiation case is the best choice for the two-step polymerization.

The weight fraction of AA-type MBP (*W*_AA_) is given by Equation (20) and its variation with *q* is shown in Figure 4B. When *q*_1_ and *q*_2_ are low, *W*_AA_ is small. Only when *q*_1_ and *q*_2_ are close to unity, *W*_AA_ increases rapidly. AA-type MBP formed in the first-step polymerization can be ignored in the two-step polymerization if monomer B is added when *q*_1_ and *q*_2_ are less than unity.

#### 3.5.2. *DP*_n,co_ of MBcP


(21)
$${DP}_{n,co}=\frac{2}{2\bar{\phi_{B}}+\left(q_{10}+q_{20}\right)(\phi_{B}-\phi_{A})}$$


When monomer A is replaced by monomer B at the time when the extent of initiation is *q*_10_ and *q*_20_, the second LRAsCP of B occurs. The *DP*_n,co_ of total MBcP is predicted by Equation (21), assuming the complete initiation of the residual function groups of BFIs in the second polymerization. The variation tendency of *DP*_n,co_ is shown in Figure 5. The theoretical analysis suggests that the termination factors of two monomers, *ϕ*_A_ and *ϕ*_B_, are the key parameters affecting *DP*_n,co_. The combination of longer polymerization of the monomer with larger *ϕ* and shorter polymerization of the monomer with smaller *ϕ* is the best way to produce MBcP with large *DP*_n,co_. For example, when styrene is first polymerized, MMA (*ϕ*_B_ = 0.33) is recommended to be added when *q*_10_ and *q*_20_ are close to unity. (Figure 5A) If the polymerization sequence of the two monomers is opposite, styrene is suggested to be added as early as possible (Figure 5B).

In summary, the structural parameters of MBPs are determined by *q* and *ϕ* according to theoretical analysis of LRAsCP. The value of *q* is determined by the initiation rate and polymerization time. *Φ* is determined by the nature of the monomer and is the key factor to constructing the multiblock structure.

## 4. Experimental Study

Two initiators, I_12_ and I_2_, shown in Figure 3, were prepared from commercially available two benzylbromides I_11_ and I_1_ by published methods [43]. I_11_ is a symmetrical dibromide corresponding to *s*-initiation case, and I_12_ is an unsymmetrical dibromide corresponding to *c*-initiation case.

### 4.1. Homopolymerization of Styrene

The styrene homopolymerization initiated by I_11_ and I_12_ were conducted under the same conditions. The monomer conversion, the number-average molar mass (*M*_n_) and its distribution of all MBPs measured by GPC are shown in Figure 6 and Appendix A.

The conversion of styrene increased with the polymerization time for both BFIs. *M*_n_ gradually increased with polymerization time before 3 h and accelerated afterwards. The variation tendency of *M*_n_ of total MBP prepared by two BFIs suggest that the coupling of macroradicals occurred in LRAsCP. Accelerated growth of *M*_n_ of total MBP after 3 h reflects the stepwise construction of the multiblock architecture. The *M*_n_ of MBPs prepared by I_11_ (*s*-initiation case) grows slightly faster than those obtained by I_12_ in the early period of polymerization, which is consistent with the tendency predicted by Equations (14)–(16) and Figure 3D. When I_12_ was used as initiator, *M*_n_ of total MBP did not remain constant at the early stage of polymerization, which suggests that LRAsCP initiated by I_12_ is not exactly *c*-initiation case, but a case between *s*-initiation and *c*-initiation. Although I_12_ is not a typical *c*-initiation case, its ester moiety introduces cleavable sites into MBPs, making the resultant MBPs degradable.


(22)
$$M_{n}=\frac{M_{n,0}}{q\left(2-q-\phi_{A}\right)}\;(s\text{-}\mathrm{initiation})$$



(23)
$$M_{n}=\left\{\begin{array}{c} \frac{M_{n,0}}{q\left(2-\phi_{A}\right)},\;\;\;\;q\leq 0.5\\ \frac{M_{n,0}}{\left(1-q\phi_{A}\right)},\;\;\;\;q>0.5 \end{array}\right.\;(c\text{-}\mathrm{initiation})$$



(24)
$$M_{n,0}=\frac{A_{0}Cm_{A}}{X_{0}}$$


The *M*_n_ values of total MBP for *s*-initiation and *c*-initiation cases are further derived from Equations (17)–(22) and (23), respectively. The ratio, *A*_0_*Cm*_A_/*X*_0_, in Equation (17) is the molar mass of polymer upon one chain per BFI, which is defined as the apparent molar mass (*M*_n,0_) and is given by Equation (24). Since the parameters *A*_0_, *X*_0_, *C*, *m*_A_ and *ϕ*_A_ are available, the extent of initiation of the two cases at different times can be solved by Equations (22)–(24), which are given by Equations (25)–(27), respectively.


(25)
$$q=\frac{\left(2-\phi_{A}\right)\pm \sqrt{{\left(2-\phi_{A}\right)}^{2}-4R}}{2}\;(s\text{-}\mathrm{initiation})$$



(26)
$$q=\left\{\begin{array}{c} \frac{R}{\left(2-\phi_{A}\right)},\;\;\;\;\;q\leq 0.5\\ \frac{1-R}{\phi_{A}},\;\;\;\;\;\;\;\;\;\;\;q>0.5 \end{array}\right.\;(c\text{-}\mathrm{initiation})$$



(27)
$$M_{n,0}=\frac{A_{0}Cm_{A}}{X_{0}}$$


Based on the feed ratio, the monomer conversion and *M*_n_ of product prepared by I_11_, the variation of *q* with the polymerization time estimated by Equation (25) is shown by Figure 7. It can be found that *q* of I_11_ for *s*-initiation case increased linearly with polymerization time because the polymerization is promoted by Mn_2_(CO)_10_/visible light. The decay of *q* when it was over 0.8 is due to the high monomer conversion and the gel effect observed in the polymerization [42].
(28)$$N_{K}=\frac{2-\phi_{A}}{2-\phi_{A}-q}\;(s\text{-}\mathrm{initiation})$$
(29)$$N_{K,GPC}=\frac{M_{n}}{M_{n}^{'}}$$

As shown in Figure 1 and Appendix A, MBP is a chain composed of several segmental chains linked by moiety X. The average number of blocks per MBP (*N*_K_) for the *s*-initiation case can be predicted by Equation (28) derived from Equation (18). The molar mass of the block is equal to the molar mass of the polymer generated by I_1_ ($M_{n}^{'}$) in conventional radical polymerization under the same polymerization conditions if the kinetics chain length (*ν*) generated from two initiators are supposed to be the same. Therefore, *N*_K_ can be estimated by the ratio of *M*_n_ to $M_{n}^{'}$ measured by GPC, which is denoted as *N*_K,GPC_ in Equation (29). As shown in Figure 8, the predicted *N*_K_ according to Equation (28) and its measured value according to Equation (29) are in good agreement.

On the other hand, the relationship between *ν* and $M_{n}^{'}$ in conventional polymerization is given by Equation (30) [42]. Therefore, *ν* can be quantified by the polymerization of monomer with known *ϕ* initiated by I_1_.(30)$$M_{n}^{'}=\frac{\nu }{1-0.5\phi_{A}}\mathrm{or}\;\nu ={(1-0.5\phi_{A})M}_{n}^{'}$$

As shown in Appendix A, the relationship between *DP*_n_ and the number of kinetic chains per MBP (*N_ν_*) varies with its type due to the different terminal units. If *ν* generated by BFI is the same as that by MFI under the same polymerization conditions, the relationship between *DP*_n_ of MBP and *ν* is deduced as(31)$${DP}_{n,GPC}=\frac{M_{n}}{2\nu }+\left(0.5F_{IA}+F_{IAI}\right)=\frac{M_{n}}{{(2-\phi_{A})M}_{n}^{'}}+\left(0.5F_{IA}+F_{IAI}\right)$$(32)$${DP}_{n,GPC}=\frac{N_{K,GPC}}{\left(2-\phi_{A}\right)}+\frac{1-q}{2-\phi_{A}-q}\;(s\text{-}\mathrm{initiation})$$
(33)$${DP}_{n}=\frac{2-q}{2-\phi_{A}-q}\;(s\text{-}\mathrm{initiation})$$

Equation (31) can be used to experimentally estimate the *DP*_n_ of MBP based on *M*_n_, $M_{n}^{'}$, *F*_IA_ and *F*_IAI_, which is denoted as *DP*_n,GPC_. For the *s*-initiation case, Equation (31) is reduced to Equation (32) and Equation (16) is reduced to Equation (33). *DP*_n,GPC_ of total MBP prepared by I_11_ experimentally estimated by Equation (32) at various values of *q* is shown in Figure 8, which fits well with *DP*_n_ predicted by Equation (33).

### 4.2. Two-Step LRAsCP

Two-step LRAsCP was conducted in two ways. One is one-pot polymerization via the addition of comonomer at a certain time. LRAsCP under the same polymerization condition as Run 4 in Appendix A was conducted. After 4 h, the polymerization was suspended by turning off the light and half of the polymer solution was collected and precipitated. The resultant sample was denoted as MBP-1 (*M*_n_ = 9.4 KDa, Ð = 2.0 in Figure 9 and Appendix A). MMA was added to the remaining half of the solution, and the second LRAsCP started by turning on the light for an additional 190 min. Sample MBcP-1 (*M*_n_ = 34.7 KDa, Ð = 5.1 in Figure 9 and Appendix A) composed of 81 mol% styrene was obtained. This is in situ two-step LRAsCP and the second LRAsCP is the copolymerization of styrene and MMA resulting in St-MMA copolymer block. The obtained MBP-1 contains both IA- and IAI-type MBPs and was used as macro-initiator for the copolymerization of styrene and butyl methacrylate (BMA) for 4 h. Sample MBcP-2 (*M*_n_ = 38.6 KDa, Ð = 5.0 in Figure 9 and Appendix A) composed of 89 mol% styrene was obtained. The improved molar masses of MBcP-1 and MBcP-2 compared with that of MBP-1 demonstrates that LRAsCP continues in the second-step polymerizations in two different ways. MBcP-2 is composed of 75 wt% polystyrene block and 25 wt% styrene-BMA (49/51, mol/mol) copolymer block and the glass transition temperature (*T*_g_) of 53 °C was detected by DSC, which is different from the *T*_g_ of PS (100 °C) and PBMA (21 °C). This confirms that a copolymer block was formed during the second-step LRAsCP.

### 4.3. Determination of ϕ of Different Monomers

The termination mode in terms of *k*_t,c_/(*k*_t,c_ + *k*_t,d_) is a key parameter in radical polymerization kinetics. Several experimental methods, such as isotopic labeling of the initiator [44], transformation of initiator fragments after polymerization [45], gelation technique [46,47] and MALDI analysis of polymer [48], have been proposed to determine the termination mode. These methods are based on quantitative analysis of the chain end, which is still difficult because the concentration of initiator fragments is low. Since the *N*_K_ of MBP produced in LRAsCP is determined by *ϕ*, it is possible to acquire the value of *ϕ* in LRAsCP. Replacing *N*_K_ in Equation (28) with *N*_K,GPC_ in Equation (29) yields Equation (34).(34)$$q=(1-N_{K,GPC}^{-1})(2-\phi_{A})$$

If the same *q* is achieved under the same polymerization conditions, Equation (34) can be applied to measure the factor *ϕ* of any polymerization based on *N*_K,GPC_ and known *ϕ* of styrene. The polymerization of styrene, MMA and BMA initiated by I_11_ and I_1_ were conducted at the same conditions for 3 h and the polymerization results are given in Appendix A. The number-average molar mass of the six samples prepared by I_11_ (*M*_n_) and I_1_ ($M_{n}^{'}$) results in different *N*_K,GPC_. With known *ϕ* of styrene (0.93) [38], the calculated value of *ϕ* of MMA is 0.33 according to Equation (34), which is the same as that determined by the measurement of gelation point in NLRAsCP of MMA [38]. The calculated *ϕ* of BMA is −0.25, which is obviously unreasonable. This is probably due to the gelation effect caused by the high monomer conversion and the chain transfer reaction to the polymer. Although the factor *ϕ* is not the same as *k*_t,c_/(*k*_t,c_ + *k*_t,d_), LRAsCP provides an easy method of quantitative determination of the contribution of the coupling reaction in the termination step of radical polymerization independent of the initiator fragment.

## 5. Conclusions

LRAsCP initiated by BFI and its kinetics modal have been proposed. Theoretical analysis of LRAsCP provides the quantitative relationships between some key parameters of MBP, such as the molar fraction of various MBPs, *DP*_n_, *M*_n_, *N*_K_ with *q* and *ϕ*. The *s*-initiation case of LRAsCP of styrene initiated by **I_11_**/Mn_2_(CO)_10_/visible light were intensively investigated. The experimental results fit well with the theoretical predictions, which approve the stepwise construction of the multiblock structure of MBP. MBcP composed of two types of blocks, PS block and P(St-MMA/BMA) block, can be produced by two-step LRAsCP. Furthermore, LRAsCP can be applied to the measurement of *ϕ* of radicals when *ϕ* of one specific monomer is known. The current results demonstrate that the introduction of BFI to conventional radical polymerization generates a new polymerization strategy, which is a facile method to construct the multiblock structure. Since radical polymerization is tolerant to various monomers, LRAsCP can be extended to other monomers that are difficult to be polymerized by living and controlled radical polymerizations.

## 6. Experimental Section

Two initiators, I_2_ and I_12_, shown in Figure 2 were prepared by the reaction between I_1_ or I_11_ and 2-bromoisobutanoic acid by published methods [43]. The structure of two compounds was confirmed by NMR, HRMS. The homopolymerization of styrene and copolymerization of styrene with MMA and BMA promoted by Mn_2_(CO)_10_/visible light were conducted. MBPs were characterized by GPC, and MBcPs were characterized by GPC and NMR. Full descriptions of the methods and analytical data for the compounds and polymer used and generated in the study are given in the Appendix A.

## Data Availability

Data are available within the article and Appendix A.

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
