# Peer review of "Linear Radical Additions-Coupling Polymerization (LRAsCP): Model, Experiment and Application"

_polymers, 2025, doi:10.3390/polym17060741_

Round 1
Reviewer 1 Report
Comments and Suggestions for Authors
The submitted manuscript describes radical addition-coupling polymerization and copolymerization of vinyl monomers (styrene, methyl metacrylate MMA, butyl metacrylate BMA) initiated by bifunctional initiator (mainly symmetrical dibromo-p-xylene in combination with Mn2(CO)10 ). The polymers/copolymers prepared were characterized by appropriate methods (GPC, NMR). The elaborated theoretical part showing relations between kinetic phenomena (like extent of initiation – q, and termination factor ϕ) and macroscopic chain characterization (like degree of polymerization) is involved (mainly as ESI). However, the correspondence between ESI and the main text is not perfect (equation numbering, missing references in the main text), which does not makes the manuscript friendly to the reader.
I have the following questions and remarks to the manuscript:
- The authors speak about formation of multiblock polymers. Well, in the case of homopolymerization, the polymer chain is divided in to blocks by molecules of initiator. However, in the case of copolymerization, there are no blocks consisting of individual monomers. In this sense some formulation in Introduction (esp. Scheme 1) are slightly misleading. Moreover, a possibility of chain cleavage in the place of initiator molecule destroy any block character.
- The introduction of ϕ = 0.93 for styrene and 0.33 for MMA on page 6 should be accompanied by proper literature reference.
- The author should explain how copolymer composition was calculated from NMR spectra (what signals were used, what is the signal a´ in Fig. S3).
- Comments to Table S4: - explain how conversion (wt%) was calculated in the case of copolymer, was it conversion of styrene or MMA? Moreover, there is discrepancy between text and table in conversion for MBcP-1. Explain how quantities Fst,1, Fst,2 and W were calculated.
- GPC traces (Fig. S4, Fig. 9) shows a bimodal distribution in some cases. Could you comment it.
- The determination of parameters q and ϕ from experimental data (Table S5) seems to me questionable. The same value of q for homopolymerization and two copolymerization experiments are speculative and therefore ϕ values for MMA and BMA are not trustful.
- Formal errors: p.15 Experimental Scheme 2 should be Scheme 3, in ESI p.17 line 4 Table S1 should be Table S2.
In summary, the manuscript bring new data about radical polymerization with bifunctional initiator. I recommend its publication after appropriate revision.
Reviewer 2 Report
Comments and Suggestions for Authors
- The Introduction section should be rewritten. Please add a text, describing the main aims of the paper, and the tasks. which will be fulfilled.
- The authors should correct the literature according to journal requirements. Please ad DOIs.
Reviewer 3 Report
Comments and Suggestions for Authors
The manuscript presents a novel polymerization strategy using a bifunctional initiator (BFI) to achieve multiblock polymerization via Linear Radical Additions-Coupling Polymerization (LRAsCP). The work provides both theoretical and experimental insights, demonstrating the feasibility of this method in polymer synthesis. While the study is interesting and relevant, several aspects require improvement in terms of scientific clarity, grammatical accuracy, and overall readability.
Clarity and Scientific Justification:
The manuscript lacks a clear and concise explanation of the novelty and significance of LRAsCP compared to existing polymerization strategies. The introduction mentions different approaches to multiblock polymer synthesis, but it would benefit from a stronger comparative analysis that highlights why this new method is advantageous.
Some theoretical explanations, particularly in the kinetics section, are dense and difficult to follow. Simplifying the descriptions or adding explanatory notes would enhance reader comprehension.
Experimental Methodology:
The section detailing the polymerization reactions lacks some specifics regarding reaction conditions. Important parameters such as pressure, temperature variations, and detailed procedural steps should be clearly stated to ensure reproducibility.
The characterization of the synthesized polymers could be more robust. While GPC (gel permeation chromatography) and DSC (Differential Scanning Calorimetry) data are provided, additional supporting techniques (e.g., FTIR) could reinforce structural claims.
Language and Grammar Issues
The manuscript contains several typographical and grammatical errors that should be addressed to improve readability and professionalism. Some notable spelling mistakes include:
"sysmmtrical" should be "symmetrical"
"correponding" should be "corresponding"
"intitiaon" should be "initiation"
"archtechture" should be "architecture"
"convesrion" should be "conversion"
These errors appear throughout the text and may affect the clarity of the content. A thorough proofreading is strongly recommended to ensure accuracy and coherence.
